# Absence of item origin bias on a Brazilian interinstitutional Progress Test examination: A pooled analysis of items approach

Pedro Tadao Hamamoto Filho[1]*, Maria de Lourdes Marmorato Botta Hafner[2],
Zilda Maria Tosta Ribeiro[2], Alba Regina de Abreu Lima[3], Leandro Arthur Diehl[4],
Neide Tomimura Costa[4], Maria Cristina de Andrade[5], Samira Yarak[5],
Patrícia Moretti Rehder[6], Júlio César Moriguti[7], Angélica Maria Bicudo[6]

1 UNESP, Universidade Estadual Paulista, Botucatu, São Paul, Brazil, 2 FAMEMA, Faculdade de Medicina de Marília, Marília, São Paul, Brazil, 3 FAMERP, Faculdade de Medicina de São José do Rio Preto, São José do Rio Preto, São Paul, Brazil, 4 UEL, Universidade Estadual de Londrina, Londrina, Paraná, Brazil, 5 UNIFESP, Universidade Federal de São Paulo, São Paulo, São Paul, Brazil, 6 UNICAMP, Universidade Estadual de Campinas, Campinas, São Paul, Brazil, 7 USP, Universidade de São Paulo, Ribeirão Preto, São Paul, Brazil

* pedro.hamamoto@unesp.br

## Abstract

### Background

It has been proposed that the school origin of items for cross-institutional Progress Tests (PTs) may introduce a bias in favour of students from the same school, posing a potential threat to the validity and reliability of PT results and cross-institutional comparisons. The aim of this study was to examine whether origin bias is present in a Brazilian cross-institutional PT examination.

### Methods

This study conducted a cross-sectional analysis of seven schools affiliated with the oldest PT consortium in Brazil, utilising a pooled analysis of differences in students' performance concerning self and non-self items. A proportional meta-analysis of the items' rate differences and confidence intervals with random effects was performed, providing an odds ratio (OR) for self and non-self items. Differences between the two groups of items were assessed by scrutinising whether the OR and 95% confidence intervals overlapped.

### Results

The findings indicated no discernible differences in psychometric indices based on the school responsible for item creation. Three schools consistently demonstrate superior performance on items authored by their faculty, however, these they also excelled on non-self items. Furthermore, an overlap in the 95% confidence intervals for both self and non-self items was observed across all seven schools.

**Data availability statement:** All relevant data are within the paper and its Supporting Information files.

**Funding:** Pedro Tadao Hamamoto Filho and Angélica Maria Bicudo have received an award from the National Board of Medical Examiners (PA, PA, USA). GRANT_NUMBER: Proposal LAG5-2020. Pedro Tadao Hamamoto Filho is supported by Conselho Nacional de Desenvolvimento Científico e Tecnológico (CNPq). GRANT_NUMBER: 313047/2023-5. The funders had no role in study design, data collection and analysis, decision to publish, or preparation of the manuscript.

**Competing interests:** The authors have declared that no competing interests exist.

## Conclusions

In contrast to prior reports, this study revealed the absence of origin bias, suggesting that adoption of best practices in blueprinting, item writing, and editing may have played a role in mitigating such bias.

## Introduction

The Progress Test (PT) stands as a widely adopted assessment tool used by medical schools worldwide to evaluate students' knowledge accumulation throughout their undergraduate program years [1–3]. Its integration into assessment programs yields several advantages, particularly in terms of valuable feedback for students, faculty members, and institutions [4–6]. The validity and reliability of the PT hinge upon the implementation of sound practices in test construction, administration, thorough analysis and review of results, and input from all stakeholders [7].

PT examinations are designed at the final-year level, theoretically allowing for curriculum and cross-institutional comparisons, provided that these curricula and institutions share similar educational goals despite potential differences in methodological approaches [8,9]. However, the effectiveness of PT can be compromised by various factors, including flaws in item writing, inclusion of irrelevant constructs, use of imprecise terms, and heterogeneity in test difficulty [7,10–12].

Another potential source of bias affecting PT results is the endogeneity effect, commonly referred to as "*origin bias*". Muijtjens et al. (2007) have explicitly defined origin bias as the phenomenon in which *"the origin of items introduced bias in favour of students from the same school as the item producers"* [13]. The underlying argument posits that students from a specific school are more likely to achieve better performance on items written by faculty members from their own school.

Inter-institutional PT offers the advantage of cost-sharing and expertise collaboration between schools. However, if origin bias is present, the fairness of assessment results would be compromised, hindering meaningful cross-comparisons. Since the original study by Muijtjens et al., there has been a notable absence of investigation into the presence of item origin bias in PT examinations. This study aims to fill the gap by examining whether origin bias is present and determining any significant differences between schools in a Brazilian cross-institutional PT examination.

## Methods

### Settings

This study took place within the most traditional Brazilian Consortium for PT, comprising nine public schools: Faculdade de Medicina de Marília (FAMEMA), Faculdade de Medicina de São José do Rio Preto (FAMERP), Universidade Federal de São Carlos (UFSCAR), Universidade Federal de São Paulo (UNIFESP), Universidade Estadual de Campinas (UNICAMP), Universidade de São Paulo (USP—Bauru and Ribeirão Preto campi), and Universidade Estadual Paulista (UNESP) in São Paulo State, and Universidade Estadual de Londrina (UEL) in Paraná State [14].

This consortium conducts biannual examinations for all students (approximately 4500) spanning the first to sixth undergraduate years. The exam comprises 120 multiple-choice questions structured around a fixed blueprint encompassing various areas, disciplines, and themes. Initially, the blueprint was distributed evenly across six equal areas (basic sciences, internal medicine, paediatrics, surgery, obstetrics, gynaecology, and public health) in adherence to national legislation for medical residency selection. However, this division led to an imbalance between areas and low reliability, prompting a blueprint modification. The current structure allocates percentages as follows: internal medicine (28.3% of items), paediatrics (19.7%), surgery (19.7%), obstetrics and gynaecology (16.7%), and public health (16.7%) [15].

During the test construction, the Coordination Committee assigns specific requests for each blueprint content. For example, if the blueprint focuses on acute coronary syndrome, the Coordination Committee can establish different requests for each examination, such as the diagnosis of unstable angina, initial treatment of acute myocardial infarction, or electrocardiogram interpretation. These requests are disseminated to faculty members from the nine participating schools. Consequently, for each blueprint content, up to nine items may be generated, with the area subcommittee selecting the best from the nine to be included in the final test. In subsequent tests, a different request from the same blueprint is provided.

## Study design and statistical analysis

For this study, data were extracted from the 6[th] (final) year undergraduate students across seven schools within the consortium. This sampling choice was made considering that the PT is designed at the final-year level of knowledge. Two schools were excluded because of the limited number of items incorporated into the test. The details of students and written items for each school are shown in Table 1.

First, we conducted an evaluation to determine whether the psychometric properties differed according to their school of origin. The difficulty index, representing the percentage of students with incorrect answers, and the discrimination index denoting the difference in the percentage of correct answers between 27% of the high performers and 27% of the low performers in the test, were computed. Differences between mean difficulty and discrimination indices were checked using one-way ANOVA with statistical significance set at $p < 5\%$, and analyses were executed using SPSS software for Mac Book (Statistical Package for Social Sciences, v. 24.0, IBM Corp, Armonk, NY, United States).

For each item, we compared the rate of correct answers between schools that authored the item and other schools. A pooled analysis of the items, as previously described [16], involved calculating the rate of correct answers for each item along with a corresponding 95% confidence interval (CI). A proportional meta-analysis of the items' rate differences and confidence intervals with random effects, was performed. The intervention group comprised students from the same school as the item author, whereas the control group consisted of other students. Consequently, each school had a final odds ratio (OR) for self-items, following the same procedure for non-self items. Heterogeneity between the rates of each item was determined using $I^2$ statistics [17]. Ultimately, a statistical difference between self and non-self items was

**Table 1. Number of 6[th] year students of participating schools.**

| School | Students |
| --- | --- |
| A | 76 |
| B | 72 |
| C | 76 |
| D | 78 |
| E | 83 |
| F | 74 |
| G | 100 |

established if their combined 95% confidence intervals did not overlap [18]. These analyses were carried out using Med-Calc for Windows, version 19.4 (MedCalc Software, Ostend, Belgium) with statistical significance was set at $p < 5\%$.

## Ethical considerations

As our study involved secondary data, with no individual students' identification (i.e., analyses were conducted at the item level, not individual level), approval from the institutional review board was deemed unnecessary, in accordance with national legislation governing research ethics involving human subjects.

Moreover, no school was disclosed in the presentation of results in accordance with the consortium's code of conduct. To prevent rankings between participating schools, each school was granted access only to the overall results and their specific outcomes for any analysis, whether for research purposes or otherwise. This approach aligns with ethical considerations to maintain confidentiality and equitable treatment among the participating institutions.

## Results

To verify that the psychometric quality of items was not different across the schools, we analysed the mean values of difficulty and discrimination of items according to the school. We observed no differences in the psychometric indices based on the school of origin for the items. The mean difficulty of the items averaged $0.39 \pm 0.20$, with variations from 0.33 (school E) to 0.44 (school F), demonstrating no statistically significant differences ($F = 0.68$, $p = 0.67$). Similarly, the mean discrimination indices showed no significant distinctions, with a mean of $0.38 \pm 0.13$, ranging from 0.33 (school A) to 0.42 (school C), and no statistical significance observed ($F = 0.77$, $p = 0.59$) (Fig 1).

In examining self-items, three schools exhibited an OR greater than 1.0, indicating a significantly higher answer rate for items crafted by their faculty. In other three schools, the ORs exceeded 1.0, however, the confidence intervals reached 1.0, and no statistically significant differences were detected. Conversely, only one school had an OR < 1.0, although the difference was not statistically significant (Fig 2).

The superior performance of these three schools on self-items could not be solely attributed to origin bias; it might be associated with their overall superior performance. For further exploration, we calculated the OR for non-self items. The same three schools demonstrated superior performance on non-self items (OR > 1.00, $p < 0.05$), whereas the remaining four schools displayed significantly inferior performance on non-self items (OR < 1.00, $p < 0.01$) (Table 2). The

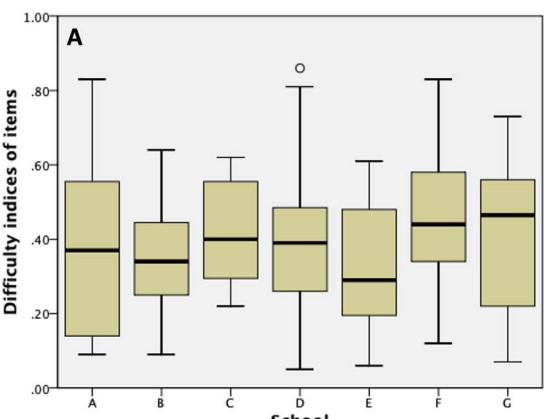
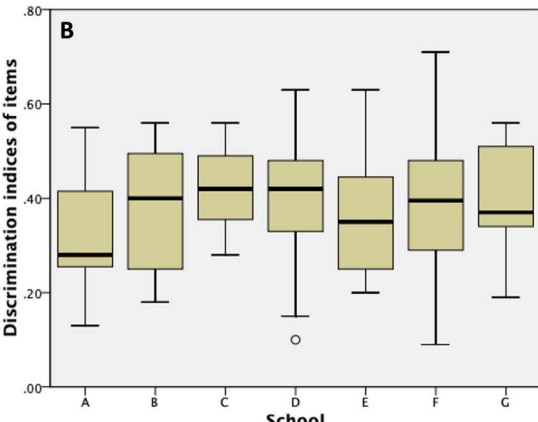

**Fig 1. Boxplot graphs illustrating the difficulty (A) and discrimination (B) indices of items based on the school origin.** No statistically significant differences were observed for both indices, highlighting the consistency in psychometric properties across the schools involved in the study.

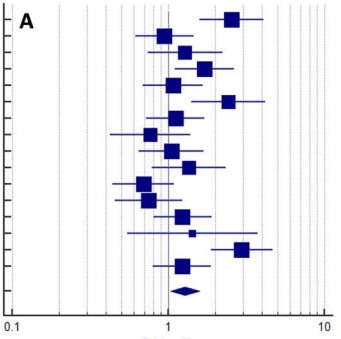 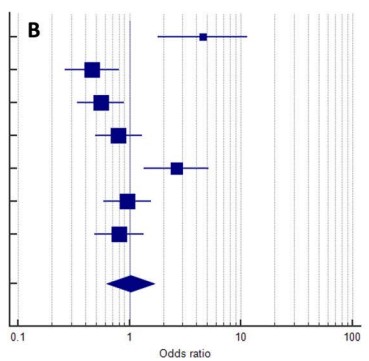 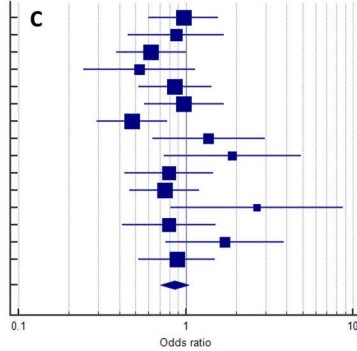

**Fig 2. Illustration of the proportional meta-analysis of pooled items, where each horizontal line corresponds to an item.** The squares represent the odds ratio, and the confidence intervals are depicted by the horizontal lines associated with each OR. Squares to the right of the "1" axis indicate superior performance in correct answers for students from the school that authored the item, whereas squares to the left signify better performance by students from other schools. The diamond represents the final result of the school's OR. Panel A shows a school with superior performance on their items (final OR does not touch "1"). Panel B indicates a school with no discernible differences in performance on their items. Panel C highlights a school whose performance trends are lower, though not significantly (final OR touches "1").

**Table 2. Proportional meta-analysis of right answer rate differences between self and non-self items.**

| School | Self items | | Non-self items | |
|---|---|---|---|---|
| | OR [95%CI], p-value | I², p-value | OR [95%CI], p-value | I², p-value |
| A | 0.86 [0.71–1.04], p=0.12 | 30.8%, p=0.12 | **0.86 [0.80–0.93], p<0.01** | 47.3%, p<0.01 |
| B | **1.22 [1.01–1.47], p<0.05** | 17.1%, p=0.28 | **1.25 [1.16–1.35], p<0.01** | 46.2%, p<0.01 |
| C | 1.02 [0.63–1.66], p=0.94 | 81.3%, p<0,01 | **0.78 [0.72–0.85], p<0.01** | 60.6%, p<0.01 |
| D | 1.08 [0.94–1.27], p=0.27 | 52.1%, p<0,01 | **0.90 [0,82−0,99], p=0.03** | 64.4%, p<0.01 |
| E | 1.01 [0.75–1.21], p=0.68 | 66.9%, p<0.01 | **0.88 [0.81–0.95], p<0, 01** | 56.9%, p<0.01 |
| F | **1.14 [1.01–1.28], p=0.03** | 0.0%, p=0.60 | **1.16 [1.07–1.27], p<0.01** | 52.1%, p<0.01 |
| G | **1.28 [1.03–1.59], p=0.03** | 67.6%, p<0.01 | **1.20 [1.11–1.29], p<0.01** | 56.1%, p<0.01 |

Note: Significant odds ratios (ORs) are in bold.

heterogeneity of rates varied from moderate to high for all but two analyses, where schools B and F exhibited low heterogeneity for self-items, with similar rates across items. The Q-test results for the heterogeneity analysis are presented in the S1 Data.

Despite the differences in performance for non-self items, there was an overlap in the combined confidence intervals for all seven schools (Fig 3), which means that the differences in students' performance between self and non-self items were not statistically significant.

## Discussion

The students' differential backgrounds were identified as a source of variability in different examinations. For example, instructor changes (staff turnover) in a Dutch school of economics caused significant variations in students' grades, as well as pass and fail rates [19]. Other studies have shown that socioeconomic variables have a significant effect on students' mean scores [20–22]. However, few studies have addressed the origins of test items as a source of variation in student performance. In the context of PT, this study represents the second attempt, to the best of our knowledge, to explore the role of origin bias in influencing students' performance.

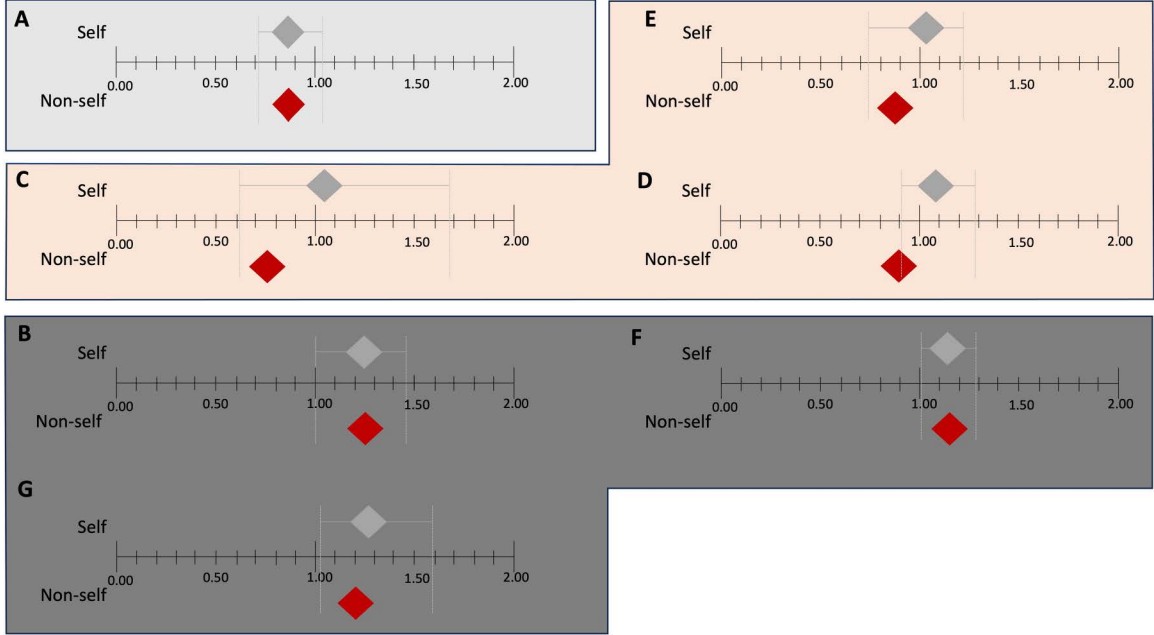

**Fig 3. Demonstrating the overlap of combined confidence intervals, the upper diamonds represent the school's performance on self items, whereas the lower diamonds correspond to the school's performance on non-self items.** The 95% confidence intervals (CI) of the upper diamonds are displayed inferiorly, whereas the CI of lower diamonds is not represented (out of scale). Across all schools, there is a noticeable overlap of CIs, indicating that the differences are not statistically significant. The schools were categorised based on the diamond's position. School A had an OR < 1.0 for both self and non-self items. Schools C, D, and E had an OR > 1.0 (though with $p > 0.05$) for self items and an OR < 1.0 (with $p < 0.01$) for non self items. Schools B, F, and G had an OR > 1.0 for both, self and non-self items ($p < 0.05$).

In contrast to the findings of Muijtjens et al. [9], our study did not reveal origin bias influencing the PT results across different institutions. Muijtjens et al. [9] demonstrated a consistent superiority in the performance of students from Maastricht University compared to those from Nijmegen, particularly on items written by Maastricht staff. This trend was especially pronounced from the 2nd to the 5th undergraduate years. They employed the *Dscore*, which gauges the difference in item difficulty between student groups, and incorporated a model with independent variables such as undergraduate year and university [13].

Criticism of their model revolves around potential biases arising from curriculum exposure discrepancies and random fluctuations. Additionally, cross-institutional comparisons might foster undesirable competitive dynamics among students and institutions [23].

In our investigation, we adopted a distinct method, focusing on the odds ratio of the percentage of correct answers for each item, rather than relying on the difference in percentages between comparison groups. Unlike Muijtjens et al. [9], we deliberately avoided performing pairwise comparisons between schools and refrained from identifying them. In our methodology, the control group for each school encompassed all other participating schools. Notably, we did not identify any origin bias that would account for differential performance among students from various institutions. Our method offers the advantage of handling sparse data while focusing on aggregate item-level effects, without requiring access to individual student performance. This feature is particularly relevant for preserving data privacy, especially in benchmarking exercises involving multiple schools. Moreover, the high heterogeneity rates on $I^2$ analyses suggest a non-uniform pattern of students' responses across the exam items, indicating high variability between schools and supporting the absence of origin bias.

The three schools (B, F, and G) whose students demonstrated superior performance on items written by their faculty also achieved the best overall performance. Although concerns may arise regarding schools C, D, and E, where students exhibited better performance on self items than on non-self items, these schools more likely underperformed consistently in the entire test, rather than specifically in the subset of non-self items, compared to B, F, and G.

This absence of item origin bias in our study can be attributed to the conscientious efforts of the Coordination Committee in adhering to the best practices for item writing. A relevant experiment conducted by Bertoni et al. in Australia involved the identification and correction of flaws in multiple-choice questions used in finance exams. Following corrections and exam re-administration, they observed enhanced clarity for students, accompanied by an increase in correct answers [24]. This underscores the substantial impact that frequent errors in item writing can have on students' overall performance [7].

Our PT consortium is actively involved in enhancing item-writing practices by conducting workshops for school faculties, adhering to international guidelines for effective assessment [25,26]. Despite these efforts, the Coordination Committee receives numerous flawed items, which may carry unconscious endogenic flags, for PT composition [27]. Our belief is that a meticulous review of items and the selection of the best item for each blueprint request contribute to the creation of uniform tests with standardised items. We recommend that schools utilizing PT not only adopt and implement a blueprint but also explicitly define the desired expectation for item writing in each blueprint component.

However, our study is not without limitations. First, we present data from a single test with a limited number of items and this cross-sectional dataset may have failed to capture dynamic aspects of item origin bias. To draw more robust and definitive conclusions, regular monitoring of item origin and differential performance based on schools for each item is crucial. Second, our conclusions are specific to 6th year students, and generalizing them to other undergraduate years should be approached with caution. Third, the moderate-to-high heterogeneity ($I^2$) in most analyses suggest the potential influence of random fluctuations across items, which cannot be entirely ruled out.

Despite these limitations, our study offers valuable insights into origin bias as a significant factor influencing variation in student performance. This overlooked concern poses a threat to the validity and reliability of common assessment tools in medical education. Furthermore, the proportion meta-analysis of pooled items adds a complementary tool to the existing set, enhancing benchmark assessments. Although we cannot rule out the possibility that methodological limitations—such as the limited power of a single cross-sectional dataset—may have contributed to the observed absence of origin bias, we believe that the adoption of best practices in PT construction more likely reflects a genuine improvement in fairness by mitigating such bias.

## Conclusions

This study found no evidence of origin bias in a progress test examination administered by a Brazilian cross-institutional consortium of medical schools. The adoption of best practices in blueprinting, item writing, and test editing may have contributed to minimizing such bias. As the use of progress test continues to expand globally, monitoring origin bias is important to enhance the validity and comparability of test results.

## Supporting information

**S1 Data. The dataset includes the number of correct answers for each item, organized by the school that developed the item, along with a comparison to the performance of students from other schools.**
(XLSX)

## Author contributions

**Conceptualization:** Pedro Tadao Hamamoto Filho, Júlio César Moriguti, Angélica Maria Bicudo.

**Data curation:** Pedro Tadao Hamamoto Filho, Maria de Lourdes Marmorato Botta Hafner, Zilda Maria Tosta Ribeiro, Alba Regina de Abreu Lima, Leandro Arthur Diehl, Neide Tomimura Costa, Maria Cristina de Andrade, Samira Yarak, Patrícia Moretti Rehder, Angélica Maria Bicudo.

**Formal analysis:** Pedro Tadao Hamamoto Filho.

**Funding acquisition:** Pedro Tadao Hamamoto Filho, Angélica Maria Bicudo.

**Investigation:** Pedro Tadao Hamamoto Filho.

**Methodology:** Pedro Tadao Hamamoto Filho.

**Project administration:** Angélica Maria Bicudo.

**Supervision:** Zilda Maria Tosta Ribeiro, Alba Regina de Abreu Lima, Leandro Arthur Diehl, Neide Tomimura Costa, Maria Cristina de Andrade, Patrícia Moretti Rehder, Júlio César Moriguti, Angélica Maria Bicudo.

**Validation:** Maria de Lourdes Marmorato Botta Hafner, Zilda Maria Tosta Ribeiro, Alba Regina de Abreu Lima, Leandro Arthur Diehl, Neide Tomimura Costa, Maria Cristina de Andrade, Samira Yarak, Júlio César Moriguti, Angélica Maria Bicudo.

**Visualization:** Samira Yarak, Patrícia Moretti Rehder, Júlio César Moriguti, Angélica Maria Bicudo.

**Writing – original draft:** Pedro Tadao Hamamoto Filho.

**Writing – review & editing:** Maria de Lourdes Marmorato Botta Hafner, Zilda Maria Tosta Ribeiro, Alba Regina de Abreu Lima, Leandro Arthur Diehl, Neide Tomimura Costa, Maria Cristina de Andrade, Samira Yarak, Patrícia Moretti Rehder, Júlio César Moriguti, Angélica Maria Bicudo.

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
