## [Decision Letter · Decision Letter 0]

23 Apr 2025

Dear Dr. Hamamoto Filho,

Thank you for submitting your manuscript to PLOS ONE. After careful consideration, we feel that it has merit but does not fully meet PLOS ONE’s publication criteria as it currently stands. Therefore, we invite you to submit a revised version of the manuscript that addresses the points raised during the review process.

**Please answer all reviewers comments and provide a point-by-point response with revised manuscript.**

We look forward to receiving your revised manuscript.

Kind regards,

Mohammad Mofatteh, PhD, MPH, MSc, PGCert, BSc (Hons), MB BCh (c)

Academic Editor

PLOS ONE

**Journal Requirements:**

Please ensure that your manuscript meets PLOS ONE's style requirements, including those for file naming. The PLOS ONE style templates can be found at https://journals.plos.org/plosone/s/file?id=wjVg/PLOSOne_formatting_sample_main_body.pdf and https://journals.plos.org/plosone/s/file?id=ba62/PLOSOne_formatting_sample_title_authors_affiliations.pdf 2. We note that the grant information you provided in the ‘Funding Information’ and ‘Financial Disclosure’ sections do not match.  When you resubmit, please ensure that you provide the correct grant numbers for the awards you received for your study in the ‘Funding Information’ section. 3. Thank you for stating the following financial disclosure: Pedro Tadao Hamamoto Filho and Angélica Maria Bicudo have received an award from the National Board of Medical Examiners (PA, PA, USA). GRANT_NUMBER: Proposal LAG5-2020.  Please state what role the funders took in the study.  If the funders had no role, please state: "The funders had no role in study design, data collection and analysis, decision to publish, or preparation of the manuscript." If this statement is not correct you must amend it as needed. Please include this amended Role of Funder statement in your cover letter; we will change the online submission form on your behalf. 4. We note that you have indicated that there are restrictions to data sharing for this study. For studies involving human research participant data or other sensitive data, we encourage authors to share de-identified or anonymized data. However, when data cannot be publicly shared for ethical reasons, we allow authors to make their data sets available upon request. For information on unacceptable data access restrictions, please see http://journals.plos.org/plosone/s/data-availability#loc-unacceptable-data-access-restrictions.  Before we proceed with your manuscript, please address the following prompts: a) If there are ethical or legal restrictions on sharing a de-identified data set, please explain them in detail (e.g., data contain potentially identifying or sensitive patient information, data are owned by a third-party organization, etc.) and who has imposed them (e.g., a Research Ethics Committee or Institutional Review Board, etc.). Please also provide contact information for a data access committee, ethics committee, or other institutional body to which data requests may be sent. b) If there are no restrictions, please upload the minimal anonymized data set necessary to replicate your study findings to a stable, public repository and provide us with the relevant URLs, DOIs, or accession numbers. Please see http://www.bmj.com/content/340/bmj.c181.long for guidelines on how to de-identify and prepare clinical data for publication. For a list of recommended repositories, please see https://journals.plos.org/plosone/s/recommended-repositories. You also have the option of uploading the data as Supporting Information files, but we would recommend depositing data directly to a data repository if possible. Please update your Data Availability statement in the submission form accordingly.

**Additional Editor Comments:**

Please answer all reviewers comments and provide a point-by-point response with revised manuscript.

Reviewers' comments:

Reviewer's Responses to Questions

**Comments to the Author**

1. Is the manuscript technically sound, and do the data support the conclusions?

Reviewer #1: Yes

Reviewer #2: Partly

2. Has the statistical analysis been performed appropriately and rigorously?

Reviewer #1: Yes

Reviewer #2: Yes

3. Have the authors made all data underlying the findings in their manuscript fully available?

Reviewer #1: No

Reviewer #2: Yes

4. Is the manuscript presented in an intelligible fashion and written in standard English?

Reviewer #1: No

Reviewer #2: Yes

**Reviewer #1:**  Dear Author,

Thank you for the opportunity to review this manuscript. Below are the comments on my review:

1. The subject matter of the article is relevant to be published in the journal. However, it needs minor revisions.

2. The title clearly reflects the subject matter of the article. However, it could mention that the study was conducted in Brazil, to better define the research.

3. The abstract and keywords provide good information about the article. The subject matter, the objective of the research, the methodology used, the main results and conclusions are mentioned. However, the abstract fails to mention the objective of the research.

4. Check whether all authors cited in the article are in the manuscript's references, and vice versa.

5. The article has a few problems with clarity and coherence in its language, which need to be revised. The Introduction needs to make the objective of the research clearer. The methodology is clear and highlights the data generation locations, data collection, method used, study participants and data analysis method. However, it is necessary to present the research conclusions, which are intertwined with the Discussion section of the manuscript.

6. There is articulation between the theme and the theoretical basis. There is a good dialogue with the international scientific literature that addresses this subject.

7. There is data analysis and coherence in the argumentation. Although there are no categories of data analysis, they are organized and analyzed, according to the method used, in dialogue with the scientific literature consulted.

8. The bibliography used is adequate and up-to-date in the article.

This is an interesting study. However, I hope these comments help improve your manuscript. Congratulations on your research.

**Reviewer #2: ** 1.The manuscript states: "This consortium conducts biannual examinations for all students (approximately 4500) spanning the first to sixth undergraduate years." However, the analysis focuses solely on 6th-year students. If the PT is designed at the final-year level and thus cannot be extended to earlier years, the authors should explicitly justify this sampling choice.

2.The use of a single-test, cross-sectional dataset may fail to capture dynamic aspects of item origin bias (e.g., temporal trends or interactions with student cohorts). The authors should discuss how this limitation might affect their findings and consider incorporating longitudinal data (e.g., multiple test administrations) to verify result robustness.

3.While the study controls for item origin, student performance may also be influenced by institutional factors (e.g., curriculum focus, teaching quality, or student demographics). The authors should address whether these were accounted for (e.g., via sensitivity analyses or covariates). Additionally, the superior performance of Schools B/F/G could stem from non-bias factors (e.g., selective admissions or targeted training). Exploring these possibilities would enrich the discussion.

4.The rationale for choosing pooled analysis over alternative methods (e.g., multilevel modeling) should be clarified. A brief explanation of its advantages (e.g., handling sparse data, aggregating item-level effects) would aid readers in evaluating the approach.

5.The random-effects model reports moderate-to-high heterogeneity (I²). To quantify this further, the authors should report Q-test p-values and explore potential sources. This would help assess whether heterogeneity undermines the "no bias" conclusion.

6. The statement "no discernible differences in psychometric indices based on the school responsible for item creation" requires deeper contextualization. The authors should:

Contrast their null result with prior positive findings (e.g., Mujijtjens et al.), highlighting methodological or contextual differences.

Discuss whether the absence of bias reflects true fairness or methodological constraints (e.g., limited power).

Emphasize how their consortium’s practices (e.g., blueprint adherence, faculty training) might mitigate bias, offering actionable insights for other institutions.

**Do you want your identity to be public for this peer review?** For information about this choice, including consent withdrawal, please see our Privacy Policy

Reviewer #1: **Yes: ** Gustavo Cunha de Araujo

Reviewer #2: No

---

## [Author Response · Author response to Decision Letter 1]

2 May 2025

RESPONSES TO THE REVIEWERS’ COMMENTS

REVIEWER #1:

Dear Author,

Thank you for the opportunity to review this manuscript. Below are the comments on my review:

1. The subject matter of the article is relevant to be published in the journal. However, it needs minor revisions.

We thank the reviewer for the thoughtful comments that certainly improved the quality of our manuscript.

2. The title clearly reflects the subject matter of the article. However, it could mention that the study was conducted in Brazil, to better define the research.

We added “Brazil” in the title.

3. The abstract and keywords provide good information about the article. The subject matter, the objective of the research, the methodology used, the main results and conclusions are mentioned. However, the abstract fails to mention the objective of the research.

We agree that the aim was missing. We added it to the Abstract.

4. Check whether all authors cited in the article are in the manuscript's references, and vice versa.

Everything is ok with this regard.

5. The article has a few problems with clarity and coherence in its language, which need to be revised. The Introduction needs to make the objective of the research clearer. The methodology is clear and highlights the data generation locations, data collection, method used, study participants and data analysis method. However, it is necessary to present the research conclusions, which are intertwined with the Discussion section of the manuscript.

We checked English Language. Also, we stated the objective at the end of Introduction. The Conclusions are presented now in a separate section.

6. There is articulation between the theme and the theoretical basis. There is a good dialogue with the international scientific literature that addresses this subject.

Thank you.

7. There is data analysis and coherence in the argumentation. Although there are no categories of data analysis, they are organized and analyzed, according to the method used, in dialogue with the scientific literature consulted.

Thank you.

8. The bibliography used is adequate and up-to-date in the article.

Ok.

This is an interesting study. However, I hope these comments help improve your manuscript. Congratulations on your research.

Thank you for appreciating our study and for taking the time to kindly review our manuscript.

REVIEWER #2:

1.The manuscript states: "This consortium conducts biannual examinations for all students (approximately 4500) spanning the first to sixth undergraduate years." However, the analysis focuses solely on 6th-year students. If the PT is designed at the final-year level and thus cannot be extended to earlier years, the authors should explicitly justify this sampling choice.

Yes, your observation is absolutely correct. We have justified our choice accordingly on the Methods section.

2.The use of a single-test, cross-sectional dataset may fail to capture dynamic aspects of item origin bias (e.g., temporal trends or interactions with student cohorts). The authors should discuss how this limitation might affect their findings and consider incorporating longitudinal data (e.g., multiple test administrations) to verify result robustness.

We have discussed the limitation of using a single-test database – we now reinforced that it really may have failed to provide more robust data. We agree with the reviewer and, even though we cannot overcome this limitation in the present study, our group is adopting longitudinal efforts to verify origin bias throughout all progress test examinations. We hope that we can present the generated data on upcoming studies.

3.While the study controls for item origin, student performance may also be influenced by institutional factors (e.g., curriculum focus, teaching quality, or student demographics). The authors should address whether these were accounted for (e.g., via sensitivity analyses or covariates). Additionally, the superior performance of Schools B/F/G could stem from non-bias factors (e.g., selective admissions or targeted training). Exploring these possibilities would enrich the discussion.

We agree with the reviewer that probably there are many factors influencing the superior performance of schools B, F, and G. However, as we have stated on Ethical Considerations section, no school was disclosed in the presentation of results in accordance with the consortium’s code of conduct. It means that not only the schools’ name would be published but also that one school will not know the performance of the students from others. We are quite aware that PT allows for cross-institutional benchmarking – however, these comparisons are made only on private sessions. We will take the reviewer’s comments to study the viability of investigating how the aforementioned factors may cause differences on PT scores, on the basis of the school’s directors’ approval. And finally, we understand that controlling for other institutional factors goes beyond the scope of the present investigation.

4.The rationale for choosing pooled analysis over alternative methods (e.g., multilevel modeling) should be clarified. A brief explanation of its advantages (e.g., handling sparse data, aggregating item-level effects) would aid readers in evaluating the approach.

We added this explanation on the Discussion.

5.The random-effects model reports moderate-to-high heterogeneity (I²). To quantify this further, the authors should report Q-test p-values and explore potential sources. This would help assess whether heterogeneity undermines the "no bias" conclusion.

We agree with the comment. We present the Q-test results as Supplemental Material. The values were presented beside The I2 results. However, we believe that high heterogeneity rates on I2 analyses suggest a non-uniform pattern of students’ responses across the exam items, indicating high variability between schools and supporting the absence of origin bias. In other words, a uniform pattern of correct answers would suggest a homogeneous pattern – possibly due to origin bias. We added this reasoning on the Discussion.

6. The statement "no discernible differences in psychometric indices based on the school responsible for item creation" requires deeper contextualization. The authors should:

Contrast their null result with prior positive findings (e.g., Mujijtjens et al.), highlighting methodological or contextual differences. Discuss whether the absence of bias reflects true fairness or methodological constraints (e.g., limited power). Emphasize how their consortium’s practices (e.g., blueprint adherence, faculty training) might mitigate bias, offering actionable insights for other institutions.

We added explanation with regards to the psychometric indices across the school’s items. We expanded the Discussion and the Conclusion (as suggested by Reviewer #1) emphasizing the interest of our method and our practices at the consortium.

---

## [Editor Report · Decision Letter 1]

18 May 2025

Absence of item origin bias on a Brazilian interinstitutional Progress Test examination: A pooled analysis of items approach

PONE-D-25-03875R1

Dear Dr. Hamamoto Filho,

We’re pleased to inform you that your manuscript has been judged scientifically suitable for publication and will be formally accepted for publication once it meets all outstanding technical requirements.

Kind regards,

Dr. Mohammad Mofatteh, PhD, MPH, MSc, PGCert, BSc (Hons), MB BCh BAO (c)

Academic Editor

PLOS ONE
---

## [Editor Report · Acceptance letter]

PONE-D-25-03875R1

PLOS ONE

Dear Dr. Hamamoto Filho,

I'm pleased to inform you that your manuscript has been deemed suitable for publication in PLOS ONE. Congratulations! Your manuscript is now being handed over to our production team.

Kind regards,

on behalf of

Dr. Mohammad Mofatteh

Academic Editor

PLOS ONE